# Perceptions of surgeons on surgical antibiotic prophylaxis use at an urban tertiary hospital in Tanzania

**Elizabeth E. Mmari**[1]*, **Eunice S. Pallangyo**[2], **Athar Ali**[1], **Dereck A. Kaale**[3], **Isaac H. Mawalla**[1], **Muzdalifat S. Abeid**[4]

1 Department of General Surgery, The Aga Khan University, Dar es Salaam, Tanzania, 2 Department of Nursing and Midwifery, The Aga Khan University, Dar es Salaam, Tanzania, 3 Department of Emergency Medicine, Saifee Hospital, Dar es Salaam, Tanzania, 4 Department of Obstetrics and Gynaecology, The Aga Khan University, Dar es Salaam, Tanzania

* elizabeth.e.mmari@gmail.com

## Abstract

### Background

Surgical Site Infections are a major cause of morbidity and mortality among operated patients. In spite of the accessibility of universal and national guidelines for surgical prophylaxis, recent studies surveying the present routine of prophylaxis have demonstrated over-utilization of a wide range antibacterial medication for a single patient. Few studies have shown qualitatively factors influencing this and perceptions of surgeons on surgical antibiotic prophylaxis use. Unfortunately, none of these studies have been done in Tanzania.

### Objective

To describe the perceptions of surgeons on surgical antibiotic prophylaxis use at an urban tertiary hospital.

### Methods

A qualitative study involving in-depth interviews with surgeons was conducted in English by the primary investigator. The interviews were audio-recorded and transcribed verbatim. Systematic text condensation by Malterud was used for data analysis.

### Findings

Fourteen surgeons and obstetrics and gynaecologists participated. Their perceptions were summarized into three main categories: Inadequate data to support practice; one who sees the patient decides the antibiotic prophylaxis; prolonged antibiotic use for fear of unknown. The participants perceived that choice of antibiotic should be based on local hospital data for bacterial resistance pattern, however the hospital guidelines and data for surgical site infection rates are unknown. Fear of getting infection and anticipating complications led to prolonged antibiotics use.

**Data Availability Statement:** De-identified data is available upon request due to ethical restrictions imposed by The Aga Khan University Ethical review committee concerning participant privacy. Gagiel

Ketto, gadiel.ketto@aku.edu is the current Librarian and researchers can get in touch with him to request data.

**Funding:** EM received funding from The Aga Khan University as part of research requirement. It was a research allowance, not a grant. https://www.aku.edu/iedea/Pages/home.aspx The funders had no role in study design, data collection and analysis, decision to publish, or preparation of manuscript.

**Competing interests:** No authors have competing interests

## Conclusion

The study provides an understanding of surgical antibiotic prophylaxis use and its implementation challenges. This was partly expressed by unavailability of local data and guidelines to enhance practice. To improve this, there is a need of guidelines that incorporates local resistance surveillance data and enhanced antibiotic stewardship programmes. A strong consideration should be placed into ways to combat the fears of surgeons for complications, as these significantly affect the current practise with use of surgical antibiotic prophylaxis.

## Background

The use of surgical antibiotics prophylaxis (SAP) is recommended in prevention and reduction of incidence of surgical site infections (SSI). SAP is the use of antibiotics for prevention of SSI before or during a surgical procedure and is usually given to patients who undergo some clean procedures or clean-contaminated procedures [1]. The Centre for Disease Control (CDC) defines a clean wound as an incision in which no inflammation is encountered in a surgical procedure, without a break in sterile technique, and during which the respiratory, alimentary and genitourinary tracts are not entered while a clean-contaminated wound, has an incision through which the respiratory, alimentary or genitourinary tract is entered under controlled conditions but with no contamination encountered [1]. The rates of SSI range from 2–5% in clean procedures to 8–10% in clean contaminated procedures which may increase based on the degree of wound contamination. A meta-analysis by Stijn et al, demonstrated the importance of timing of administration (30–60 minutes prior to incision time), selection of agent for specific microbes (narrow spectrum antibiotics) and duration of prophylaxis (single pre-operative dose or intraoperative re-dosing if indicated) in prevention of SSI [2]. Unfortunately, non-compliance to guidelines is still observed and prolonged use of antibiotics leads to an increase in SSI rates and increased antimicrobial resistance rates [3–5]. An antimicrobial resistance situation analysis in 2015 in Tanzania indicated the resistance of *Streptococcis pnemoniae* to Trimethoprim and Sulphamethoxazole had increased from 25% in 2006 to 80% in 2012. *Escherichia coli* from urinary infections showed a 90% resistance to Ampicillin and 30–50% resistance to other antibiotics. Extended-Spectrum *Beta Lactamases* (ESBL), which causes resistance to all beta lactam antibiotics was found in 25–40% of *E.coli* [6].

SSI remain a major cause of morbidity and mortality among operated patients and represent about a fifth of all healthcare- associated infections (HCAI) [7]. The economic impact of nosocomial infections in countries with fewer resources is far greater than in high income countries due to the larger number of infections and smaller health budgets [8,9]. Inappropriate use of antibiotics increases the total cost of health care, and its impact is worse in low income countries [9].

The studies below show the importance of compliance to antibiotic guidelines, with a decrease in SSI rates shown with good compliance. LaBove et al conducted a study in 2016 in Australia about office based elective surgical procedures [10]. This study indicated decreased SSI rates by 0.36% when recommended guidelines were followed. The study showed that in 99.6% of the time, antibiotics were administered within 60 minutes of incision time. This was in contrast with in-hospital patients where SSI rates of 3.7% were documented in the same hospital. Regardless of the difference in SSI rates in these two different settings, the SSI rates

documented in this study show a lower rate in high income countries compared to lower income countries. A study in Switzerland assessing for optimal timing and choice of SAP on 21007 cardiac patients revealed an SSI rate of 5.5% and administration of SAP within 30 minutes of incision was associated with lower SSI rates [11].

In contrast to the above studies, poor compliance with guidelines is associated with increased SSI rates in below studies. A study in Italy was conducted to assess the appropriate use of antibiotics in 382 patients admitted in the hospital. This study revealed that only 18.1% of the patients received appropriate antibiotics based on choice, dose and duration of antibiotics and only 53.4% received antibiotics within 60 minutes of incision time [4]. Similarly, a multicentre study in Brazilian hospitals showed high rates of SSI with full compliance of all SAP guidelines in only 10% of surgeries. They recommended need for innovative stewardship approach to improve adherence to SAP guidelines [12].

In lower income countries, the SSI rates frequently are reported higher than 10% (in USA it is estimated that the SSI rate is about 3%) [8]. There is no information showing the rate of SSIs in Tanzania. Though different studies have been done in independent health facilities showing higher rates than in high income countries. A study conducted in rural Tanzania in Ifakara to show antimicrobial prophylaxis usage to prevent SSI showed a SSI rate of 22% with patients receiving prophylaxis after incision time [5]. Another study in Kilimanjaro Christian Medical Centre (KCMC) carried out to identify the incidence of SSI in 388 patients showed a SAP usage of 87% of the patients, with 19.4% developing SSI and having antimicrobial resistance [13]. In addition to these studies, Akoko in Muhimbili National Referral Hospital (MNRH) conducted a study evaluating risk factors of developing SSI [14]. They showed an SSI rate in 35.6% of patients, although there was no attribute to show how use of antibiotics affected the rates of SSI in the study and a higher correlation to HIV patients [14]. These studies done in Tanzania all similarly show an increased rate of SSI, in contrast to documented rates from the CDC. They also show inappropriate use of SAP, whether in choice, timing or duration of SAP.

Ethnographic studies have been done to understand the factors affecting SAP use. Charani et al 2017 showed ward rounds were the most important part in decision making when it comes to choice and duration of antibiotics [15]. He also noted that surgeons considered surgical operations to be an important part of their management and hence antibiotic decision making was left to junior doctors. Jennifer et al observed that antibiotics were given low priority in comparison to the surgical operation theatre requirements and there existed a lack of trust in the antibiotic guidelines to protect the surgeon from managing patient infections when they do occur [16]. A long duration of SAP was prescribed in belief to be beneficial to the patient and improvisation gave the surgeons a sense of relief that there is a lower risk of developing infection. A systematic review by Ng *et al* showed a lack of awareness of guidelines that govern SAP use and Initial training influenced decision making in SAP use [17]. Personal preference and influence of colleagues were among factors that impaired adherence to guidelines.

In spite of the accessibility of universal and national guidelines for SAP [18], recent studies surveying the present routine of prophylaxis have demonstrated that, overutilization of antimicrobial medications, unnecessary utilization of broad-spectrum antibiotics and wrong planning and span are still a frequent finding and hazardous [19,20]. In Tanzania, the specialist surgical workforce in 2016 was 0.46 per 100,000 population [21]. High SSI rates and inappropriate use of antibiotics is documented quantitatively [5,22,23]. There are no qualitative studies done in Tanzania looking into perceptions of surgeons on SAP use. This study aimed to explore the perceptions of surgeons on SAP use.

## Study design

### Study setting

A qualitative study using in-depth individual interviews was conducted at the Aga Khan hospital, Dar es Salaam. It is a 170-bed private tertiary hospital with surgical department consisting of general surgery, urology, orthopaedics, otolaryngology and neurosurgery. Approximately 15 surgeries are carried out per day and 5400 per year. The current practice of antibiotic use involves decision making that occurs during out-patient clinic and ward rounds, where choice, dosage and route of antibiotic administration is decided. The antibiotics are administered in the pre-operative patient waiting bay before the start of surgery. The duration of antibiotic is usually decided intra or post-operative during ward rounds.

The Hospital has an antimicrobial stewardship program from surgery, pharmacy and microbiology departments. It has demonstrated an improvement in choice of SAP after its initiation in its data base. There is still documented increased duration of antibiotic use, inadequate timing, and antibiotics given in clean surgeries. Furthermore, bacterial drug resistance has been documented by the hospital laboratory to the routinely used antibiotics in most of the cultured microbes.

Ethical clearance was obtained from the Aga Khan University Ethical Research Committee, and the Aga Khan hospital. The individual participants have given written informed consent (as outlined in PLOS consent form) to publish these case details.

### Study participants

Participants were recruited from surgery and obstetrics and gynaecology department regardless of their super specialty. Purposive sampling technique was used to determine participants who would have valuable information. An invitation letter by email was sent to 40 surgeons who use SAP, 17 accepted, 23 did not respond. In order to ensure rich and diverse data, surgeons of both genders, varying in age and work experience were recruited. The Table 1 below summarizes the participants' characteristics.

### Data collection

Surgeons who accepted to participate received written informed consent and underwent in-depth individual interviews using a pre-structured interview guide. The interview guide was modified based on initial interviews to improve clarity and flow. It focused on understanding of SAP, experience with SSI, decision making of SAP and current practice with regards to SAP. A total of 14 interviews were conducted by primary researcher in English in a quiet place at convenience of surgeon and lasted less than 45 minutes. All interviews were audio-recorded using a tape recorder with permission from the participants. Field notes were written during and immediately after the interview. No more than one interview was carried out per day to ensure enough time was given for the interviewer to transcribe and scrutinize data prior to the next interview.

### Data analysis

The data was analysed using the systematic text condensation, a qualitative approach by Malterud [24]. This was found to be a suitable method for analysing and systematic presentation of the manifest content of the surgeons' perceptions. The main steps of analysis involved familiarization with the content through multiple readings and developing themes and meaning units in which the primary investigator was engaged. The meaning units were then coded and grouped into categories and subcategories which were labelled at manifest level. All Authors

**Table 1. Demographic description of participants.**

| Variable | Sub-groups | N(14) |
|---|---|---|
| Age | 18–29 | 1 |
| | 30–39 | 2 |
| | 40–49 | 8 |
| | >50 | 3 |
| Sex | Male | 12 |
| | Female | 2 |
| Speciality | Surgical resident | 2 |
| | Obstetrics and gynaecologist | 3 |
| | General surgeon | 2 |
| | Urologist | 2 |
| | Otolaryngologist | 2 |
| | Neurosurgeon | 1 |
| | Ophthalmologist | 1 |
| | Orthopaedic surgeon | 1 |
| Work experience | <2 years | 3 |
| | 2–10 years | 11 |
| Work hours | Full time | 9 |
| | Part time | 5 |

N = number of participants.

read the transcripts and familiarized with the content until consensus of categories was reached. Three categories were defined and named. This was followed by a write up of the final analysis from data [24]. A summary of analysis process is shown in Table 2 below.

## Findings

The perceptions of participants with regard to SAP use were summarised into three categories: Inadequate hospital data to support practice, antibiotics are prolonged for fear of unknown and the one who sees the patient should decide SAP. The quotations by participants are indented and labelled by responder number. A summary of major categories and subcategories is given in Table 3 below.

### Inadequate hospital data to support practice

Multiple participants expressed the lack of hospital-based data to show bacterial patterns that would guide SAP use. This was shown together with unawareness of existence of hospital SAP guidelines and even those who were aware of the guidelines expressed that guidelines were not easily available when needed and hence could not be referred to frequently. Mostly participants who were working part-time were unaware of hospital guidelines.

**Table 2. Systematic text condensation method of analysis; an example of extract from the data analysis.**

| Meaning unit | Code | Category |
|---|---|---|
| From the pharmacy, it is ok they deal with drugs but . . . they don't know the patient and they don't know how we are dealing with this patient in theatre. It is ok to give suggestions but they cannot make the final decision | Pharmacists cannot decide but suggest, they don't see patient | The one who sees the patient decides SAP |

**Table 3. A summary of categories and sub categories.**

| Categories | Sub categories |
| --- | --- |
| Inadequate hospital data to support practice | No hospital data for bacterial resistance patterns |
| | Hospital guidelines are not well known |
| | Unknown hospital data for SSI |
| The one who sees patient decides SAP | Surgeon decides SAP |
| | Team members decide SAP |
| Prolonged SAP for fear of unknown | Fear of getting infection |
| | Anticipating complications |

**No hospital data for bacteria resistance patterns.** Participants noted that appropriate choice of SAP should be made based on local patterns of bacteria and their susceptibility to antimicrobials. The hospital currently uses AMS laboratory committee to follow up bacteria susceptibility patterns and there isn't a systemised data collection method in place to get adequate data to support practice. Hence international guidelines were used to make hospital guidelines. In discussion about choice of SAP some participants mentioned this below.

*. . .yeah it is good to follow guidelines, but you need to have your own data which supports what you are using because the bacteria pattern is different in different areas. . . .But guidelines are there to help you out but they cannot dictate, so you need local data to support you more. (Responder 3)*

*So, we need to know at the Aga Khan hospital when we are doing maybe perianal surgeries, which antibiotics seem to be more effective and which antibiotics seem to be more resistant. So, if we get infection, we should make sure that we do culture and sensitivity. (Responder 9)*

**Hospital guidelines are not well-known.** Few participants were not aware of availability of hospital guidelines with regards to surgical antibiotic prophylaxis and they reported to use other sources to guide them including WHO guidelines and literature. Some participants speculated that probably due to working as part-time specialists, not all hospital resources are availed to them. Participants who were aware of the guidelines expressed that a constant discussion with the pharmacy team with regards to availability and choice of SAP was done. Regardless of other participants being aware of hospital guidelines, they noted that the guidelines were not easily accessible when required.

*No, I have never seen guidelines at the Aga Khan Hospital. Maybe it is there. You know we are working part-time and maybe they do not show me the guidelines. (Responder 12)*

*I use WHO guidelines, there are no local guidelines for that. (Responder 7)*

*We have a guideline; it is user-friendly but it is not very well-known. I'm sure if you go there at the nurses' desk you will take maybe 5 minutes to find it. (Responder 13)*

**Inadequate hospital data for surgical site infections.** The SSI rates were reported to be few by most participants. The part time participants reported having multiple cases of SSI in other places of work. Participants attributed the causes of surgical site infection to include change in SAP when bacterial resistance occurred, patient factors like co-morbid and immunosuppression, pre-existing infection at the surgical site, environmental factors like sterility of the theatre and the wards, sterility of surgical instruments, surgical technique and tissue handling. Participants expressed that multiple factors are involved in development of SSI and it's

not determined only by use of SAP. However, participants emphasized the importance of surgical technique and sterility in the prevention of surgical site infection than relying on SAP alone. Regardless of this, there is inadequate documentation and follow-up of patients who develop SSI to determine the exact rates of patients developing SSI and hence determine the root cause.

> *For cases that we do at Aga Khan, that is not a problem. But for cases at Muhimbili especially when we do a lot of operations involved in the perineum like urethroplasty, infections occur for example patients with Fournier's gangrene, hypospadias in children, they get contaminated, so it depends on the site. For the upper tract, maybe nephrectomies, exploration, infection is not much of a big problem. (Responder 9)*

> *We do not have proper statistics in terms of surgical site infections.(Responder 13)*

> *I have seen a couple of them but not that much. And I think it's one of the devastating complications post-operative. It really doesn't depend on whether prophylactic antibiotics was given or not, but rather a degree of wound contamination during the procedure. (Responder 14)*

### The one who sees patient decides SAP

There was a difference in opinion with regards to decision making of SAP. Some participants advocated for teamwork from pharmacists, surgeon, and residents to interns while some participants expressed that only the surgeon could make the final decision about which SAP to use. Participants expressed that for the decision to be made by another person other than the surgeon, the person had to be part of the team managing the patient and should be taught prior to being allowed to make the decision. Regardless of the fact that the surgeons had to make the decision, having a new batch of intern doctors or nurses was noticed to be associated with inappropriate SAP use which indicated that decisions most often fell onto the intern doctors.

**Surgeon decides SAP.** The surgeon reviews the patient, is aware of the procedure to be carried out, knows all the factors involved in the patient and is in an informed position to decide SAP. The pharmacists do not see the patient and hence can only recommend SAP based on availability for the surgeon to decide. Having a new batch of intern doctors or nurses was noted to be associated with inappropriate surgical antibiotic prophylaxis use this could show that decision most often fell onto the intern doctors.

> *So, the surgeon should decide. Even though I know there are committees dealing with making local guidelines, and they're also international ones and everything. But when you are the surgeon, you are the one who is going deep into the patient. (Responder 4)*

> *Surgeon decides. So, a person in the pharmacy cannot just sit and decide that you should give this antibiotic. The pharmacist can sit in a meeting but include the surgeons, include everyone else, people who are working hand-in-hand with that patient in seeing the outcomes and then make the decision. (Responder 5)*

**Team members decide SAP.** Other participants advocated for a team decision-making process involving all stakeholders that are responsible for the patient. The surgeon should not be expected to follow up on SAP that each patient is receiving but it should be the responsibility of anyone who has seen the patient. The surgeons are not always aware of the SAP given to the patient and usually the resident or intern doctors that review the patient decides SAP. This

would in turn explain the fact why multiple participants said having new residents or new intern doctors was associated with an increase in inappropriate use of surgical antibiotic prophylaxis. However, it was emphasized that the other team members making the decision had to be part of the team managing the patient and it was the responsibility of the surgeon to lead team members about appropriate SAP use.

> *In the current practice the surgeon is not always aware of the surgical antibiotic prophylaxis given. They would admit the patient and say "send pre-operative antibiotics". . . The resident or intern doctor on call decide SAP. (Responder 11)*

> *I think this has to be team work from the pharmacist, the Resident, the staff in the ward then the surgeon. Whoever has seen the patient and thinks that this is a different case has to inform the rest of the team, starting from the surgeon to the residents in the ward, the pharmacists. (Responder 2)*

## Prolonged antibiotics for fear of unknown

Multiple participants expressed that prolonged antibiotics duration was given for fear of wound contamination that may occur and fear of complications, sepsis or wound infection that could develop. Prolonged antibiotics were observed to be given to patients with comorbid or high-risk patients. The inability to control the theatre environment, ward sterility and home environment increased likelihood of postoperative antibiotics.

**Fear of getting infection.** Participants are well aware of the recommendations about use of a single dose of preoperative antibiotics. Regardless of this they reported giving prolonged SAP with a duration of more than 24-hours up to two weeks. This was done due to some factors like the environment not considered to be sterile enough, overcrowding in the theatres and sterility in the wards not being adequate enough. These factors were considered as beyond the control of the surgeon. Because of the above, there is always a fear of a patient developing an SSI after the surgery.

> *. . .if you think I can't even maintain the theatre environment and the crowding there and there's always a risk of infection, well then you have a fear that if you use a low antibiotic then maybe you would not be able to control infection. (Responder 3)*

> *We work in this environment, our wards are not so sterile and there is a lot of contamination everywhere, so probably it is their fear, or probably it is the ideal in our condition who knows. (Responder 6)*

**Anticipating complications.** Participants also expressed giving prolonged antibiotics in anticipation of complications that could occur, fear of wound contamination and sepsis. Prolonged antibiotics were observed to be given to patients with comorbid or high-risk patients. This would be decided depending on factors like patient having emergency procedures, intraoperative findings of necrosis on the operative site, and many more. In routine cases with no anticipated complications patients did not receive prolonged SAP.

> *. . .someone already having an emergency caesarean section with prolonged labour, so we are anticipating sepsis. These are the ones will continue giving antibiotics. (Responder 1)*

> *So, you need to see accordingly each patient, what would be the infection burden, so that you can use maybe 5 to 7 days or sometimes if you think there is a risk of prostatitis or epididymitis then that may be prolonged to 2 weeks (Responder 3)*

## Discussion

Inappropriate use of SAP is a contributing factor to the observed national antimicrobial resistance patterns [6]. Multiple factors are associated with inappropriate SAP use. In the present study, we explored the perceptions of surgeons and obstetrics and gynaecologists with regards to SAP use which emerged with three main categories.

The basic principle of antimicrobial prophylaxis in surgery is to achieve adequate serum and tissue drug levels that exceed the minimum inhibitory concentrations (MIC) for the organisms that are likely to be encountered during the operation [25]. Some factors like obesity are known to affect the MIC. One single dose of an antimicrobial agent is enough for most surgical operations without requiring an additional dose [26,27]. The duration of the surgery lasting more than the half-life of the prophylactic antibiotic can affect the tissue levels and hence second dose is required in longer procedures [25].

Prolonging SAP for fear of unknown was done in anticipation of infections and or complications that could occur to the patient. This gave the surgeons a sense of surgical "comfort" that a patient is less likely to develop an infectious complication, and even if they did, the surgeon did all they could to prevent it. A surgeon is responsible and held accountable for decisions made in management of a patient, and this drives need for risk reduction which allows for clinical autonomy. Broom et al found non concordance prolonged duration of SAP was driven by a sense of benefit for individual patient [16]. Together with this, the study showed participants lacked confidence in the guidelines to protect surgeons against adverse consequences and a fear of developing infectious complications usually resulted in prolonging of SAP. Ierano showed that fears and perceptions of risk hindered appropriate SAP prescribing even though there was awareness of how inappropriate prescribing may contribute to development of multidrug resistant organisms [28]. In the present study, similar findings were observed. The knowledge of participants with regards to SAP use may not always be the only limiting factor when it comes to appropriate SAP use. As shown in the present study, participants were aware of the recommendations but still demonstrated prolonged SAP use.

Patients with co-morbidities, or immunosuppression were considered high risk, and were more likely to be given prolonged SAP, or receive SAP in procedures that do not require SAP. Complications are part of the surgical practise and anticipating them to prevent them before they occur is considered an ideal approach. However, multiple studies have showed no difference in development of SSI rates from patients who receive the ideal SAP versus those receiving prolonged doses [27,29]. It is hence important to address the fears of surgeons by antimicrobial stewardship committees, in order to achieve improved SAP use.

In the present study, two distinct views emerged with team decision making and surgeon to make final decision. The decision-making process is critical in identifying which SAP are used by the patient during the procedure. The surgeon has thorough knowledge of their patient and is aware of the procedure to be carried out, and hence would make a more informed choice of SAP. On the other hand, a team decision is possible in the current setting as this is a teaching hospital and ward rounds involve a diverse group of professions from nurse, intern doctors, pharmacists, residents and specialists. Hence sometimes decision about SAP can be made by the team after discussions and not solely by the surgeon. Charani et al found that surgeons considered their main task to be in the operating theatre and antibiotic decision making was a secondary task [15]. Thus, this would commonly be delegated to the junior team members. This could be considered similar to the present study where some surgeons advocated for a team decision making as the surgeon was not always expected to know which antibiotics every patient was receiving.

Inappropriate SAP use was noted when new residents or new interns were recruited showing that decision making process of SAP sometimes falls up on the junior doctors and hence is

of lower priority in comparison to other surgical requirements. Patients for elective surgeries who are admitted a day prior to surgery are initially reviewed by either a resident or an intern who will discuss with surgeon about the required SAP. In some routine cases, the procedures and antibiotics are assumed to be well known by the intern doctors and residents and since they are expected to be well aware of the guidelines. In other cases, the decision is based solely on the knowledge of the resident or intern doctor admitting the patient and discussion does not occur like in emergency cases. Participants advocated for continuous education of team members to improve SAP use.

In present study, there is a lack of awareness of existing of guidelines and lack of local antimicrobial patterns to support current guidelines. Similar to the present study, Pons-Busom et al found that unawareness of guidelines was a major factor influencing SAP misuse [30]. Concurrently Kasteren et al found barriers to adherence to guidelines included a lack of awareness due to ineffective distribution of current versions of guidelines and lack of agreement by surgeons to local guidelines [31]. The Aga Khan Hospital guidelines are made in accordance to international guidelines as currently there aren't any local hospital studies to show hospital antimicrobial patterns. The development of SSI is due to multiple factors but a thorough follow up of patients to audit the current practises and causes could lead to improvement in SAP use. An Infection Control Committee is responsible for coordination, implementation and evaluation of comprehensive infection control programs. It would be an effective way for surveillance of infection but also creation of policy and procedures to ensure adequate SAP use in prevention of SSI. We advocate for further research to strengthen hospital data and make antimicrobial guidelines based on local data. We recommend best means of improving programs to contain antimicrobial resistance should involve collaboration among various specialities and all healthcare professionals to achieve shared knowledge and widespread diffusion of practice.

## Strengths and limitations

This study highlighted perceptions of surgeons from multiple specialties and participants who work in other centres and hence we had diverse group to provide rich data with regards to SAP use. The study was carried out in English and hence no meaning was lost from translation. To ensure trustworthiness, credibility was ensured by involving perceptions of surgeons from multiple specialties and involving senior residents too and hence involved a diverse group of people with rich information from different sources. Transferability was ensured by providing the context of the study with a detailed elaboration of the study setting and its participants, so as a reader may infer findings to their own situation. Dependability was ensured by review of the initial interview guide after observing of the interview process by authors to ensure the information captured was relevant to the study. Multiple authors are well conversant with qualitative studies. Conformability was ensured by conducting study while following methodological process and intricate analytical steps from different authors until a consensus was reached about findings [32].

It was a single centre study and hence given the context, results may be transferred only to a setting with similar qualities and cannot be generalised to the entire country. The present study did not involve any members of Anti-microbial stewardship committee, pharmacists, pathologists and anaesthesiologists that would have provided more information with regards to development and implementation of guidelines and SAP practice in the hospital.

## Conclusion

The study aimed to describe the perceptions of surgeons towards SAP, revealing; prolonged SAP for fear of unknown, inadequate hospital data, accessibility and unawareness of

availability of guidelines influence current practice of surgeons and the one who sees the patient decides SAP. A strong consideration should be placed into ways to combat the fears of surgeons for complications, as these significantly affect the current practise. The study shows widespread variation in the current practise with regards to SAP use and urgent intervention is required to improve practise. The AMS committee needs to assess the decision-making process to understand whom it falls upon to make a decision and hence equip them with adequate knowledge to guide their decision. Hospital guidelines are an important tool in ensuring appropriate use of SAP. Making guidelines available, or conducting frequent trainings in the use of guidelines or carrying out interventional studies with regards to SAP practise could boost the current practise.

## Supporting information

**S1 Appendix. Interview guide.**
(PDF)

## Acknowledgments

We thank the participants who accepted to take part in the study. We thank the hospital management that allowed the study to be conducted.

## Author Contributions

**Conceptualization:** Elizabeth E. Mmari, Eunice S. Pallangyo, Muzdalifat S. Abeid.

**Data curation:** Elizabeth E. Mmari, Eunice S. Pallangyo, Muzdalifat S. Abeid.

**Formal analysis:** Elizabeth E. Mmari, Eunice S. Pallangyo, Athar Ali, Dereck A. Kaale, Isaac H. Mawalla, Muzdalifat S. Abeid.

**Methodology:** Elizabeth E. Mmari, Eunice S. Pallangyo, Muzdalifat S. Abeid.

**Visualization:** Elizabeth E. Mmari, Eunice S. Pallangyo, Muzdalifat S. Abeid.

**Writing – original draft:** Elizabeth E. Mmari, Eunice S. Pallangyo, Athar Ali, Dereck A. Kaale, Isaac H. Mawalla, Muzdalifat S. Abeid.

**Writing – review & editing:** Elizabeth E. Mmari, Eunice S. Pallangyo, Athar Ali, Dereck A. Kaale, Isaac H. Mawalla, Muzdalifat S. Abeid.

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
