## [Decision Letter · Decision Letter 0]

7 Jun 2021

PONE-D-21-07406

Perceptions of surgeons on surgical antibiotic prophylaxis use at an urban tertiary hospital in Tanzania

PLOS ONE

Dear Dr. Mmari,

Thank you for submitting your manuscript to PLOS ONE. After careful consideration, we feel that it has merit but does not fully meet PLOS ONE’s publication criteria as it currently stands. Therefore, we invite you to submit a revised version of the manuscript that addresses the points raised during the review.

We look forward to receiving your revised manuscript.

Kind regards,

Leonardo Solaini, MD

Academic Editor

PLOS ONE

Journal Requirements:

2. Please include a copy of the interview guide used in the study, in both the original language and English, as Supporting Information, or include a citation if it has been published previously.

3. Please include a copy of Table 3 which you refer to in your text on page 10.

5. We note that your paper includes detailed descriptions of individual patients/participants. As per the PLOS ONE policy (http://journals.plos.org/plosone/s/submission-guidelines#loc-human-subjects-research) on papers that include identifying, or potentially identifying, information, the individual(s) or parent(s)/guardian(s) must be informed of the terms of the PLOS open-access (CC-BY) license and provide specific permission for publication of these details under the terms of this license. Please download the Consent Form for Publication in a PLOS Journal (http://journals.plos.org/plosone/s/file?id=8ce6/plos-consent-form-english.pdf). The signed consent form should not be submitted with the manuscript, but should be securely filed in the individual's case notes. Please amend the methods section and ethics statement of the manuscript to explicitly state that the patient/participant has provided consent for publication: “The individual in this manuscript has given written informed consent (as outlined in PLOS consent form) to publish these case details”.

Reviewers' comments:

Reviewer's Responses to Questions

5. Review Comments to the Author

Reviewer #1: This is an interesting study of the perception and attitude of surgeons to antibiotic prophylaxis. It is a topical issue and the findings from this type of study can be used to guage education and training of health staff in infection prevention. This study can be improved by attending to the following points:

1 Some information about the burden of SSI; resistance to antimicrobial agents; and number of surgeons in Tanzania.

2. Present a clear definition of antibiotic prophylaxis and the indications for their use (would classification).

3. The purposive sampling method is by definition a subjective and judgemental process. How representative are the views of the surgeons studied to from your institution or country? How can you avoid selection bias?

4. What is the explanation for the low assent rates for surgeons in your institution?

5. Mislabelling of Tables - Line 143, Table 1 should read Table 3.

6. Re-analyse to show how many surgeons hold the different categorical positions reported

7. Reflect in the Discussion about duration of true antibiotic prophylaxis and also about the potential role of Infection Control Committees in hospitals.

8. This study reveals widespread variation in practice that requires urgent action to correct it. Perhaps the conclusion should reflect this.

Reviewer #2: The manuscript is well written, however, there are some minor corrections or clarifications to be done. Line 80, should it be utilization of broad spectrum antibiotics. Line 92 should be ..route of antibiotic administration, line 98, add the word use so as to read increased duration of antibiotic use. Line 129 first word should be main units. Line 141, check tenses of the sentence. Line 157, who are the former participants being refered to? Line 281, a reference should be added to the first sentence in the discussion section. Line 331-332, the sentence is not clear. Line 372, the word equip has been mispelled.

In this study, descision on antimicrobials used was made inconjuction with the pharmacy, however, the role of laboratory in informing antimicrobial pattern was not discussed. In your setting, is the pharmacy responsible for antimicrobial resistance testing? Is the laboratory part of the antimicrobial committee? Could not having the opinion of the laboratory also be contributing to poor SAP use?

Reviewer #3: Interesting paper describing attitudes of surgeons about antibiotic prophylaxis and the prevention of surgical site infections. To better introduce the study, I suggest to include a first part about the impact that surgical site infections have around the world and in your region.

---

## [Author Response · Author response to Decision Letter 0]

21 Jul 2021

Reviewer 1:

This is an interesting study of the perception and attitude of surgeons to antibiotic prophylaxis. It is a topical issue and the findings from this type of study can be used to guage education and training of health staff in infection prevention. This study can be improved by attending to the following points:

Dear Reviewer, 

On behalf of all co-authors, we thank you for the valuable input and feedback on the submitted paper. Below we address the concerns raised in the review:

1 Some information about the burden of SSI; resistance to antimicrobial agents; and number of surgeons in Tanzania.

Response: We agree that further information is required to provide context. We have provided the burden of SSI in Tanzania in Page 5-6, lines 98-112: 

We have provided information on resistance to antimicrobial agents on Page 4, lines 66-71: “An antimicrobial resistance situation analysis in 2015 in Tanzania indicated the resistance of Streptococcis pnemoniae to Trimethoprim and Sulphamethoxazole had increased from 25% in 2006 to 80% in 2012. Escherichia coli from urinary infections showed a 90% resistance to Ampicillin and 30-50% resistance to other antibiotics. Extended-Spectrum Beta Lactamases (ESBL), which causes resistance to all beta lactam antibiotics was found in 25-40% of E.coli(6).”

We have included number of surgeons on Page 7 line 129-130; “In 2016, the specialist surgical workforce was 0.46 per 100,000 population” There is limited information on number of surgeons in Tanzania, and the latest available data is from data collected by Lancet Commission on Global Surgery in 2016.

2. Present a clear definition of antibiotic prophylaxis and the indications for their use (would classification).

Response: We agree with the reviewer and have elaborated further the definitions provided. Initially we had defined antibiotic prophylaxis on page 4 line 51-53 as, 

” SAP is the use of antibiotics for prevention of SSI before or during a surgical procedure and is usually given to patients who undergo some clean procedures or clean-contaminated procedures (1)” 

We have added the CDC wound classification to support this information on page 4 line 53-61 

We apologize if the definition was not well elaborated. We have changed the supporting text from “A meta-analysis by Stijn et al, demonstrated the importance of timing of administration, selection of agent for specific microbes and duration of prophylaxis in prevention of SSI (2).” 

To read as this on page 4 lines 61-64; “A meta-analysis by Stijn et al, demonstrated the importance of timing of administration (30-60 minutes prior to incision time), selection of agent for specific microbes (narrow spectrum antibiotics) and duration of prophylaxis (single pre-operative dose or intraoperative re-dosing if indicated) in prevention of SSI (2)”

3. The purposive sampling method is by definition a subjective and judgemental process. How representative are the views of the surgeons studied to from your institution or country? How can you avoid selection bias?

Response: We agree with the reviewer that purposive sampling is a subjective and judgmental process. However, the choice of purposive sampling depends on the nature and aim of the research. This study was not aimed to generate results that will be used to create generalizations pertaining to the larger population, rather a detailed description for understanding the phenomena. To fulfil this, participants who are potential for providing a rich contribution were included. The findings can be used to understand similar situations in institutions with a similar context elsewhere but not for generalization to the country. We have revised the limitations from the previous statement: “It was a single centre study and hence given the context, results may be transferred only to a setting with similar qualities” to read as this on page 23 line 432-433; “It was a single centre study and hence given the context, results may be transferred only to a setting with similar qualities and cannot be generalised to the entire country.”

The primary researcher had experience of working with the surgeons for at least four years prior, and hence the participants were well known and chosen for the potential of providing adequate information for the study. To avoid selection bias, inclusion and exclusion criterion were used to facilitate recruitment. Study participants were chosen after thorough deliberation among the multiple co-authors. During data collection, there was continuous discussion and reflection that was conducted between the co-authors. 

4. What is the explanation for the low assent rates for surgeons in your institution?

Response: We thank you for your question. The most rational explanation for low SSI rates would be because of inadequate documentation and reporting of SSI cases in the institution. Each surgeon was giving their perception of SSI rates based on their practice. There was no hospital data base that shows the true rate of SSI. In the revised manuscript Page 15-16 lines 251-253 we expressed this from the initial manuscript;

 “Regardless of this, there is inadequate documentation and follow-up of patients who develop SSI to determine the exact rates of patients developing SSI and hence determine the root cause.”

5. Mislabelling of Tables - Line 143, Table 1 should read Table 3.

Response: We agree and we have rectified the heading to Table 3 on page 13 line 201

6. Re-analyse to show how many surgeons hold the different categorical positions reported

Response: We appreciate the reviewer’s insightful suggestion to re-analyze to show how many surgeons hold the different categorical positions; however, since this is a qualitative study, re-analyzing the data as suggested would defeat the purpose of a qualitative research in which the aim is not to show how many hold the different categorical positions but rather analyze their perceptions of surgical antibiotic prophylaxis through content expressed.

7. Reflect in the Discussion about duration of true antibiotic prophylaxis and also about the potential role of Infection Control Committees in hospitals.

Response: We agree with this comment and have revised the discussion to include true duration of antibiotic prophylaxis. On page 20 Lines 345-351. 

“The basic principle of antimicrobial prophylaxis in surgery is to achieve adequate serum and tissue drug levels that exceed the minimum inhibitory concentrations (MIC) for the organisms that are likely to be encountered during the operation(22). Some factors like obesity are known to affect the MIC. One single dose of an antimicrobial agent is enough for most surgical operations without requiring an additional dose(23,24). The duration of the surgery lasting more than the half-life of the prophylactic antibiotic can affect the tissue levels and hence second dose is required in longer procedures(22).”

We have revised the discussion to include the potential role of the infection control committee on page 22-23 lines 407-412

“An Infection Control Committee is responsible for coordination, implementation and evaluation of comprehensive infection control programs. It would be an effective way for surveillance of infection but also creation of policy and procedures to ensure adequate SAP use in prevention of SSI.”

8. This study reveals widespread variation in practice that requires urgent action to correct it. Perhaps the conclusion should reflect this.

Response: We agree with the reviewers comment and have amended the manuscript to reflect this. On Page 25 line 444-446: 

“The study shows widespread variation in the current practise with regards to SAP use and urgent intervention is required to improve practise”

Reviewer #2: 

The manuscript is well written, however, there are some minor corrections or clarifications to be done. 

Dear Reviewer, 

On behalf of all co-authors, we thank you for the valuable input and feedback on the submitted paper. We would also like to thank you for the generous comments that we believe have helped improve the structure of this manuscript. Below we address the issues raised in the review:

1. Line 80, should it be utilization of broad spectrum antibiotics. 

Response: We agree and have changed, the new sentence reads as follows on page 7 line 128; “In spite of the accessibility of universal and national guidelines for SAP (16), recent studies surveying the present routine of prophylaxis have demonstrated that, overutilization of antimicrobial medications, unnecessary utilization of broad-spectrum antibiotics and wrong planning and span are still a frequent finding and hazardous”

2. Line 92 should be ..route of antibiotic administration, 

Response: We agree and have made suggested changes. New sentence on page 8 line 144 reads as; “The current practice of antibiotic use involves decision making that occurs during out-patient clinic and ward rounds, where choice, dosage and route of antibiotic administration is decided”

3. line 98, add the word use so as to read increased duration of antibiotic use. 

Response: We agree and have made changes on page 8 line 150 reads as; “There is still documented increased duration of antibiotic use, inadequate timing, and antibiotics given in clean surgeries”

4. Line 129 first word should be main units. 

Response: We thank you for your comment. However, the ‘meaning unit’ mentioned in the sentence refers to the methodology for analyzing data which comprises of analyzing data in terms of; meaning units, codes and categories

5. Line 141, check tenses of the sentence. 

Response: We thank the reviewer for pointing this out and have revised the sentence to read from this; “The quotations by participants will be indented and labelled by responder number.” To read as follows on page 13 Line 199 as; “The quotations by participants are indented and labelled by responder number.”

6. Line 157, who are the former participants being refered to? 

Response: We thank you for the observation. This was a typing error. We have removed ‘former’ and replaced it with ‘some’. Previously statement reads as; “In discussion about choice of SAP former participants mentioned this below.”

 The current sentence on Page 14 line 216 reads as: “In discussion about choice of SAP some participants mentioned this below.”

7. Line 281, a reference should be added to the first sentence in the discussion section.

Response: We thank you for this observation, a reference has been added on Page 20 Line 342

8. Line 331-332, the sentence is not clear. 

Response: We agree with this observation and have revised the sentence on Page 22 Lines 399-401 to read as; “Similar to the present study, Pons-Busom et al found that unawareness of guidelines was a major factor influencing SAP misuse”

9. Line 372, the word equip has been mispelled.

Response: We agree and have corrected the error on page24 line 448

10. In this study, descision on antimicrobials used was made inconjuction with the pharmacy, however, the role of laboratory in informing antimicrobial pattern was not discussed. In your setting, is the pharmacy responsible for antimicrobial resistance testing? Is the laboratory part of the antimicrobial committee? Could not having the opinion of the laboratory also be contributing to poor SAP use?

Response: We thank you for the observation and apologize for this information not being clear. In our setting, the laboratory is responsible for antimicrobial resistance testing and providing quarterly antibiogram. The laboratory is responsible for informing us in case of increased bacterial drug resistance. We have made amendments to this statement to reflect that. Page 8 line 151-153; “Furthermore, bacterial drug resistance has been documented by the hospital laboratory to the routinely used antibiotics in most of the cultured microbes.”

Yes, the laboratory pathologist is part of the antimicrobial committee. We have revised the statement on page 8 lines 148-149 to include the different members of the committee: “The Hospital has an antimicrobial stewardship program from surgery, pharmacy and microbiology department.”

We agree that not having the opinion of laboratory can also be a contributing factor to poor SAP use. This is reflected from the findings of the study subcategory that reads ‘No hospital data for bacterial resistance patterns’ on Page 14 Line 212-214. “The hospital currently uses the AMS laboratory committee to follow up bacteria susceptibility patters but there is no systemized data collection method in place to adequately monitor resistance patterns.” 

Reviewer #3: 

Interesting paper describing attitudes of surgeons about antibiotic prophylaxis and the prevention of surgical site infections. 

Dear Reviewer, 

On behalf of all co-authors, we thank you for your insightful and comprehensive feedback on the submitted paper. We would also like to thank you for your generous comments which we believe helped improve the paper and raise important issues for consideration.

1. To better introduce the study, I suggest to include a first part about the impact that surgical site infections have around the world and in your region.

Response: We thank you for this in put and have revised the manuscript to include the impact of SSI around the world and in our region on Page 5-6 lines 73-112.

---

## [Editor Report · Decision Letter 1]

30 Jul 2021

Perceptions of surgeons on surgical antibiotic prophylaxis use at an urban tertiary hospital in Tanzania

PONE-D-21-07406R1

Dear Dr. Mmari,

We’re pleased to inform you that your manuscript has been judged scientifically suitable for publication and will be formally accepted for publication once it meets all outstanding technical requirements.

Kind regards,

Leonardo Solaini, MD

Academic Editor

PLOS ONE

---

## [Editor Report · Acceptance letter]

18 Aug 2021

PONE-D-21-07406R1 

Perceptions of surgeons on surgical antibiotic prophylaxis use at an urban tertiary hospital in Tanzania. 

Dear Dr. Mmari:

I'm pleased to inform you that your manuscript has been deemed suitable for publication in PLOS ONE. Congratulations! Your manuscript is now with our production department. 

Kind regards, 

on behalf of

Dr. Leonardo Solaini 

Academic Editor

PLOS ONE